# Mental healthcare expenditure among adults by type of mental healthcare and its association with age and sex in the Netherlands between 2015 and 2020

**Lotte Dijkstra** [1]*, **Sinan Gülöksüz**[2,3], **Albert Batalla**[1], **Jim van Os**[1]

**1** Department of Psychiatry, University Medical Centre Utrecht, Utrecht, The Netherlands, **2** Department of Psychiatry, Yale University, New Haven, Connecticut, United States of America, **3** Department of Psychiatry and Psychology, Maastricht University Medical Centre, Maastricht, The Netherlands

\* l.g.dijkstra@umcutrecht.nl

## Abstract

We aimed to explore temporal patterns of mental healthcare expenditure over the period 2015−2020 by age and type of mental healthcare, as well as by sex. A records-based cohort study using comprehensive data from health insurers in the Netherlands at the 4-number postal code level was performed. We used cluster-weighted linear regressions to examine temporal patterns of mental health-care expenditure by age group (young: 18−34 vs old: 35−65), sex, and type of care. We examined the interaction of age with sex by adding the interaction between age, year, and sex to the model. The predicted costs were higher for ages 18−34 compared to older adults aged 35−64 for all mental healthcare costs except general practitioner mental healthcare (GP-MHC). The baseline regression coefficient for the young age group was 0.23 (95%-CI = 0.21;0.24) for all mental healthcare costs combined, 0.22 (95%-CI = 0.21;0.24) for specialized mental healthcare (SPECIALIST-MHC), 0.08 (95%-CI = 0.07;0.09) for basic mental health care (BASIC-MHC) and −0.02 (95%-CI = −0.03;-0.01) for GP-MHC. There was evidence for an interaction between age and year (p < 0.0001). Women had higher mental healthcare costs than men at all time points. We found an interaction between age, sex and year (p < 0.0001), with a stronger increase in the age-cost association SPECIALIST-MHC in women, while men showed a lower but more strongly increasing age-cost association for BASIC-MHC and GP-MHC. Our study found that while young women represent an increasing share of SPECIALIST-MHC, BASIC-MHC, and GP-MHC show a stronger rise in costs for young men. Future research should address whether this is due to an overlooked need for care among young men.

**Data availability statement:** Insurance data at the 4-number postal code level from Vektis are not publicly available because Vektis has decided to only make 3-number postal code level publicly available. Data can be requested by contacting Vektis at opendata@vektis.nl The original datasets containing the socio-economic status per postal code from the Dutch Central Bureau for Statistics are publicly available via the following links: https://www.cbs.nl/nl-nl/maatwerk/2024/24/sociaal-economische-status-per-vierci-jferige-postcodehttps://www.cbs.nl/nl-nl/maatwerk/2022/34/sociaal-economische-sta-tus-per-postcode-2014-2019 The original datasets containing urbanicity per postal code from the Dutch Central Bureau for Statistics are publicly available via the following links: https://opendata.cbs.nl/statline/portal.html?_la=nl&_catalog=CBS&tableId=83220NED&_theme=784https://opendata.cbs.nl/statline/portal.html?_la=nl&_catalog=CBS&tableId=83487NED&_theme=784https://opendata.cbs.nl/statline/portal.html?_la=nl&_catalog=CBS&tableId=83765NED&_theme=784https://opendata.cbs.nl/statline/portal.html?_la=nl&_catalog=CBS&tableId=84286NED&_theme=246https://opendata.cbs.nl/statline/portal.html?_la=nl&_catalog=CBS&tableId=84583NED&_theme=246https://opendata.cbs.nl/statline/portal.html?_la=nl&_cata-log=CBS&tableId=84799NED&_theme=246 Stata do-files used for analysis are available upon request to UMC Utrecht by contacting onderzoekdivisiehersenen@umcutrecht.nl

**Funding:** The author(s) received no specific funding for this work.

**Competing interests:** The authors have declared that no competing interests exist.

## Introduction

Mental health problems appear to be on the rise in younger adults across Europe, including in the Netherlands [1–4]. In line with this trend, we previously reported on an increasing association between mental health care expenditure and young age in the Netherlands, especially in young women [5]. For the Netherlands and looking from an occupational perspective, an increase in burn-out symptoms among younger employees and in particular young women over the period 2015–2022 has also been reported [6]. However, little has been published on how the rise in mental health problems in younger adults and young women relates to the type of mental health-care used. For the Netherlands specifically, ten Have et al. also showed an increased use of mental healthcare utilization in recent years [2]. In the Netherlands, it has also previously been shown that female gender was associated with more mental health-care use in a cohort over the period 1996–1999 [7]. Furthermore, a recent newspaper report mentioned an increase in emergency room (ER) visits for self-harm by young women over the period 2013–2022 [8]. However, detailed information on trends in the type of mental healthcare per age category and sex is lacking.

Understanding the context of the mental healthcare system is important to be able to understand changes in the types of care used over time. Especially since mental healthcare systems globally and across Europe vary considerably [9]. Health-care in the Netherlands is largely financed through an insurance-based system with managed competition, where it is mandatory by law to be insured. Most men-tal healthcare is thus also financed this way, with the exception of forensic care, costs associated with long-term clinical stays, and care for minors. Compared to for instance its neighbour Belgium, the Netherlands has fewer psychiatric beds per cap-ita but a higher number of admissions per capita [10]. The mental healthcare system in the Netherlands is organised across different levels depending on the complexity of an individual's mental healthcare needs. In the Netherlands, many general practi-tioners employ a specialised assistant who offers short-term non-medical treatment for non-complex mental problems. This is referred to as General Practitioner Mental Healthcare (GP-MHC). A general practitioner can also refer a patient to two catego-ries of mental healthcare. Basic Mental Healthcare (BASIC-MHC) offers non-medical and limited medical treatment to patients with common mental disorders that are not complex. This care is usually provided by psychologists or mental health nurses, with occasional supervision by a psychiatrist. Specialist Mental Healthcare (SPECIALIST-MHC) offers all types of treatment for all mental disorders and is intended for more complex problems. This care is provided by a wide range of professionals, but more medical personnel, such as psychiatrists, are involved compared to the BASIC-MHC. SPECIALIST-MHC can also include outreaching care in the form of home visits and inpatient care, which BASIC-MHC does not offer. SPECIALIST-MHC accounts for the bulk of mental healthcare expenditures in the Dutch system [11]. Data for comparing mental health services across Europe or internationally is scarce [9].

This study aims to explore mental health expenditure by age and sex across the different levels of the mental healthcare system in the Netherlands, which to our

knowledge has not been previously reported on. Understanding this can be valuable for service planning. We thus reported results per level of the mental healthcare system to explore differences in the type of services used. In our previous analysis, we used insurance data at the 3-number postal code level, stratified by age and sex [5]. Here, we analysed mental healthcare expenditures over the period 2015–2020 as a function of age and sex at the 4-number postal code level. As the 3-number postal code area is a relatively large geographical area and is often heterogenous, the 4-number postal code area is more suitable for examining social, economic, and geographical factors. We hypothesized that we would observe a relative increase in mental health expenditure in young adults aged 18–34 years compared to other age groups over recent years, in line with our previous report, and that this would be most pronounced for women and for SPECIALIST-MHC.

## Methods

### Dataset and measures

For this records-based cohort study, we used data from Vektis, which registers all health insurance costs nationally [12]. We previously reported on insurance data in the Netherlands [5]. The unit of analysis was the first 4-number postal code area (PC-4) out of the total 6-digit postal code area subdivided by age into age groups (18–20 years and 5-year groups from there on) and sex, meaning that each observation is a group of people living in the same postal code area of the same age group and of the same sex in the same calendar year. These data were not publicly available and provided by Vektis on the 12th of September 2022 for the years 2015–2020. Individual-level data or 6-number postal code area data were not available for privacy reasons. Units of analysis of fewer than 10 people were removed from the dataset to ensure privacy. There was no information on the reason for mental healthcare use in the database, as this would require individual level data. In line with our previous report, people over 65 years were not included to prevent age-specific effects of neuropsychiatric conditions from obscuring the results. Age was dichotomized with a cut-off of 35 years. In the absence of a universally agreed-upon definition of young adulthood, we chose this cut-off for young adulthood in line with a previous report from the Netherlands on the rise in common mental health disorders to facilitate comparability [2,13]. The dataset was merged with public datasets from the Central Bureau for Statistics (CBS) on the level of urbanicity of each postal code [14]. Urbanicity was defined by the density of addresses per square kilometre and divided into five categories ranging from low to high urbanicity by the CBS. It was also merged with data from the CBS on neighbourhood-level socioeconomic status [15]. The CBS socioeconomic status is a coefficient based on a multiple correspondence analysis of income, occupational and educational data, which were divided into quintiles for the purpose of this analysis [15].

The main outcome of interest for this analysis was the average mental health care expenditure per insured person. Costs associated with long-term (>1 year) clinical stay in a mental health facility were excluded from the analysis as this happens in a few centralised clinics and thus confounds the association with postal code level variables. Forensic care, care for minors, and social care were also excluded from this analysis because these are funded through the Department of Justice and the municipalities. Mental healthcare expenditures were subdivided into SPECIALIST-MHC, BASIC-MHC, and GP-MHC. Age was the main exposure of interest, and sex was the main effect modifier of interest. Urbanicity and socioeconomic status were the confounders.

### Procedures and analysis

Analyses were conducted using Stata version 18 [16]. Means, standard deviations, medians, and interquartile ranges for the mental healthcare costs were calculated by age group to describe the data. To assess the normality of the outcome data, histograms and extreme values were inspected. As mental healthcare costs were non-normally distributed, these were zero-skewness-log-transformed, centred, and expressed in standard deviation units for analysis of associations with the predictors. Data were checked for missingness. In the case of missing data the means and confidence intervals of the log-transformed standardized total mental healthcare costs per insured person were calculated for the observations

that had missing data and those that had no missing data separately. These calculations were used to inspect for potential associations of the main outcome with missingness. Similarly, percentages of missing data per age group were calculated to assess associations of missingness with the main exposure of interest. To assess whether urbanicity, sex, and socioeconomic status were potential confounders, means and confidence intervals of log-transformed standardized mental healthcare costs per category of urbanicity, sex, and socioeconomic status were calculated. We also made cross-tabulations of age groups and the potential confounders with percentages of the total number of insured people per cell.

A minimally adjusted linear regression of total mental healthcare costs and age, weighted for the number of insured years per unit of analysis and accounting for sex, and year, as well as the interaction between age and year, was performed. Similar regressions were conducted for SPECIALIST-MHC, BASIC-MHC, and GP-MHC costs. Next the fully adjusted models adding urbanicity and socioeconomic status to the regressions were done. All regressions accounted for clustering by 4-number postal code using cluster-robust standard errors, which allowed for the use of analytic weights. Marginal estimates for the age groups were calculated for this fully adjusted model and plotted by year to visualise these results. Additionally, the marginal effect of age on mental healthcare costs by year was estimated and plotted. Subsequently, minimally adjusted linear regressions for costs and sex weighted for the number of insured years per unit of analysis and accounting for age and year, as well as the interaction between sex and year and clustering by 4-number postal code were conducted. After this the urbanicity and socioeconomic status were again added to the model for the fully adjusted regression. Marginal estimates for sex were calculated from this model and plotted by year. Next, the three-way interaction between age, year, and sex was added to both the minimally and fully adjusted linear regression models, and the marginal effect in the fully adjusted model of age on mental healthcare costs was plotted by sex and year. Wald tests were used to assess the hypotheses in all the models.

STROBE Guidelines were used for reporting [17].

## Ethics statement

As no individual level data, but only data on an aggregated postal code level was used for this study, no ethical clearance of informed consent was needed for this study. The involved researchers had no access to identifying data from individuals.

## Results

### Descriptives

There were a total of 372468 units of analysis in the sample. For the year 2015, the sample represented a total of 99.10% of the total Dutch population aged 18–65 [18]. For 2020, this was 99.32% [19]. The average number of people per unit was 168.86, with all units of analysis combined representing a total of 62896360 person-years. The mean total mental healthcare expenditure per insured person-year in the sample was 171.42 euros (SD = 176.38), ranging from 0 euros to 10932.48 euros with a median of 134.17 euros (IQR = 55.46–238.42). Table 1 shows the mean and median costs in euros

**Table 1. Mean and median mental healthcare costs per person by age-group.**

| | 18-34 years | | 35-64 years | |
|---|---|---|---|---|
| | Mean (SD) | Median (IQR) | Mean (SD) | Median (IQR) |
| Total mental healthcare costs* | 214.60 (206.45) | 177.71 (84.46;292.71) | 150.82 (155.90) | 116.49 (46.12;211.26 |
| GP-MHC | 3.00 (3.29) | 2.23 (0.56;4.33) | 2.80 (3.18) | 2.08 (0.34;4.09 |
| BASIC-MHC | 21.93 (23.95) | 16.94 (0.96;32.20) | 15.59 (19.04) | 11.42 (0;22.99) |
| SPECIALIST-MHC | 189.68 (200.77) | 153.37 (62.50;262.01) | 132.43 (152.38) | 97.76 (27.45;188.70) |

*Costs associated with long-term (>1 year) clinical stay in a mental health facility, forensic psychiatry, or costs paid by the municipality were excluded.

per person-year by age group. All healthcare costs were non-normally distributed and right-skewed. There was missing data for 1.49% of observations for socioeconomic status, 0.09% of observations for urbanicity, and 0.03% of observations for postal code. Missingness of urbanicity, socioeconomic status and postal code appeared to be associated with total log-transformed standardised mental healthcare costs based on inspecting means and confidence intervals. Missingness of postal code level socioeconomic status was associated with age. Urbanicity, socioeconomic status, and sex appeared to be associated with total mental healthcare costs. Urbanicity and socioeconomic status were found to also be associated with age and were thus considered confounders in further analyses. Given the very low percentages of missing data, no sensitivity analyses to assess the impact of missing data were conducted.

## Mental healthcare costs and age

Over the observed period, all mental healthcare costs increased in both younger and older adults except for a drop in costs associated with BASIC-MHC between 2019 and 2020 (Fig 1). The marginal prediction for mental healthcare costs was higher for younger adults aged 18−34 compared to older adults aged 35−64 for all mental healthcare costs except GP-MHC over the observed period (Fig 1), with the baseline regression coefficient in the fully adjusted model for young age being 0.23 (95%-CI = 0.21;0.24) for the all mental healthcare costs combined, 0.22 (95%-CI = 0.21;0.24) for SPECIALIST-MHC, 0.08 (95%-CI = 0.07;0.09) for BASIC-MHC and −0.02 (95%-CI = −0.03;-0.01) for GP-MHC. Results were in the fully adjusted models were similar to those in the minimally adjusted models, though coefficients were slightly smaller in the fully adjusted models (see supplementary information S1, S4, S7, S10 in S1 File). The supplementary information shows the predicted costs by age group at each time point and both the fully adjusted and minimally adjusted full regression outputs (S1,S2, S4, S5, S7, S8, S10, S11 in S1 File).

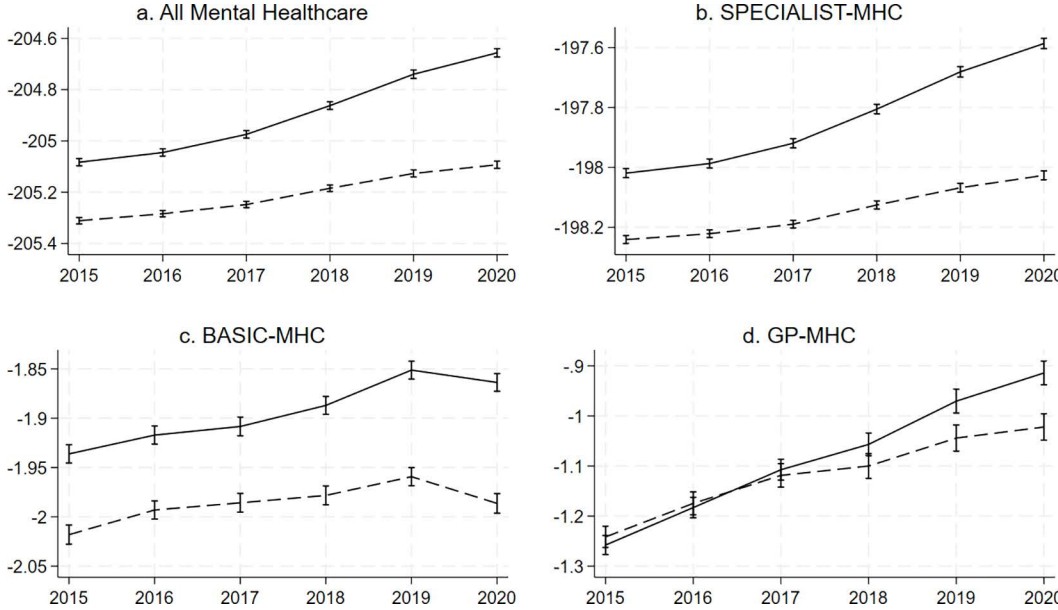

**Fig 1. Predicted log-transformed standardised costs by age group accounting for urbanicity, sex, socioeconomic status and the interaction between age and year.** The figure shows graphs of predicted log-transformed standardised costs by age group for all mental healthcare (a.) and by type of mental healthcare (b., c., d.). The X-axis shows the calendar year. The y-axis shows the log-transformed standardised costs. The uninterrupted line represents the age group 18-34 and the interrupted line represents the age group 35-64. The vertical bars intersecting the lines at each year represent confidence intervals.

The analysis showed evidence for an interaction between young age and year (p < 0.0001 for total mental healthcare costs), with the association between all mental healthcare costs and young age increasing over the observed period. The association for total mental healthcare costs in the fully adjusted model increased from 0.23 (95%-CI = 0.21;0.24) in 2015 to 0.44 (95%-CI = 0.42;0.45) in 2020. For costs associated with GP-MHC, the direction of the association changed from negative in 2015, −0.02 (95%-CI = −0.03;-0.01), to positive in 2020, 0.11 (95%-CI = 0.10;0.12) (Fig 2). The supplementary information shows the marginal estimates for the association between costs and young age in the fully adjusted models at each time point (S3, S6, S9, S12 in S1 File).

### Mental healthcare costs, age and sex

The marginal prediction for mental healthcare costs was higher for women than men at all time points and for all types of mental healthcare (Fig 3), with the baseline regression coefficient in the fully adjusted model for sex being 0.28 (95%-CI = 0.27;0.29) for all mental healthcare costs combined, 0.23 (95%-CI = 0.22;0.24) for SPECIALIST-MHC, 0.18 (95%-CI = 0.17;0.18) for BASIC-MHC and 0.43 (95%-CI = 0.42;0.44) for GP-MHC. Results were again similar in fully adjusted and minimally adjusted models (see supplementary information S13, S15, S17, S19 in S1 File). The supplementary information shows the predicted log-transformed standardised costs by sex at each time point and the full regression outputs (S13-S20 in S1 File).

Women had an overall stronger association between young age and mental healthcare costs at all time points (p < 0.0001). There was evidence for an interaction between age, calendar year, and sex for total mental healthcare costs (p < 0.0001), with women having a stronger increase in the association between younger age and total mental healthcare cost as well as SPECIALIST-MHC costs over the years, and men a stronger increase in the association between age and BASIC-MHC and age and GP-MHC costs (Fig 4). For women, the marginal estimate in the fully adjusted model for the association between age and SPECIALIST-MHC costs went from 0.35 (95%-CI = 0.33; 0.37) in 2015 to 0.65

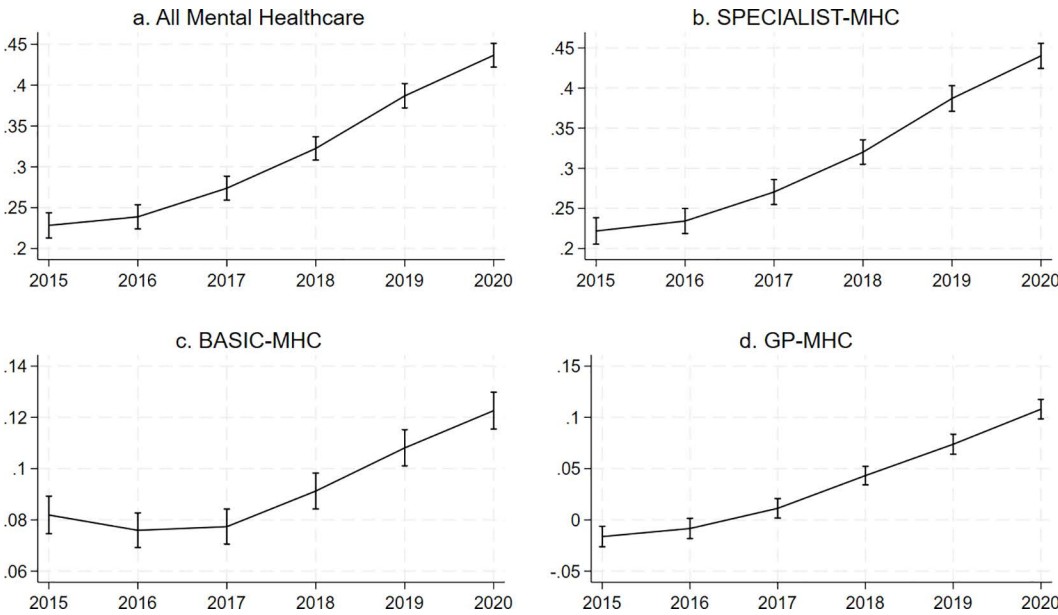

**Fig 2. Margins plots of the association between costs and young age by year accounting for urbanicity, sex, socioeconomic status and the interaction between age and year.** The figure shows graphs plotting the marginal estimates of the association between costs and young age by year for all mental healthcare (a.) and by type of mental healthcare (b., c., d.). The Y-axis shows the regression coefficient for age. The X-axis shows the calendar year. The vertical bars intersecting the lines at each year represent confidence intervals.

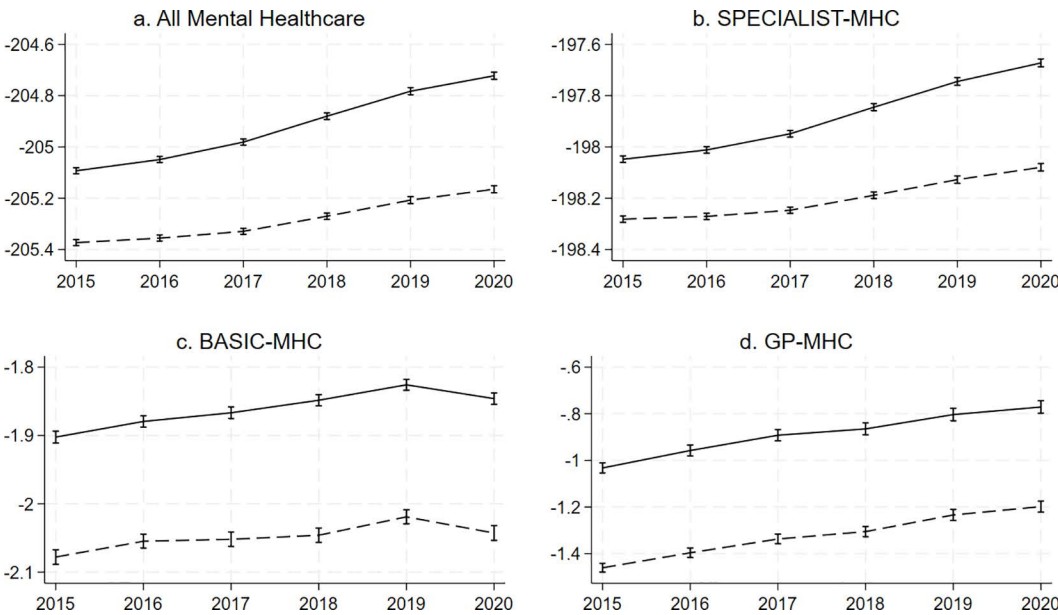

**Fig 3. Predicted log-transformed standardised costs by sex accounting for urbanicity, socioeconomic status, age, and the interaction between sex and year.** The figure shows graphs of predicted log-transformed standardised costs by sex for all mental healthcare (a.) and by type of mental healthcare (b., c., d.). The y-axis shows the log-transformed standardised costs. The X-axis shows the calendar year. The uninterrupted line represents women and the interrupted line represents men. The vertical bars intersecting the lines at each year represent confidence intervals.

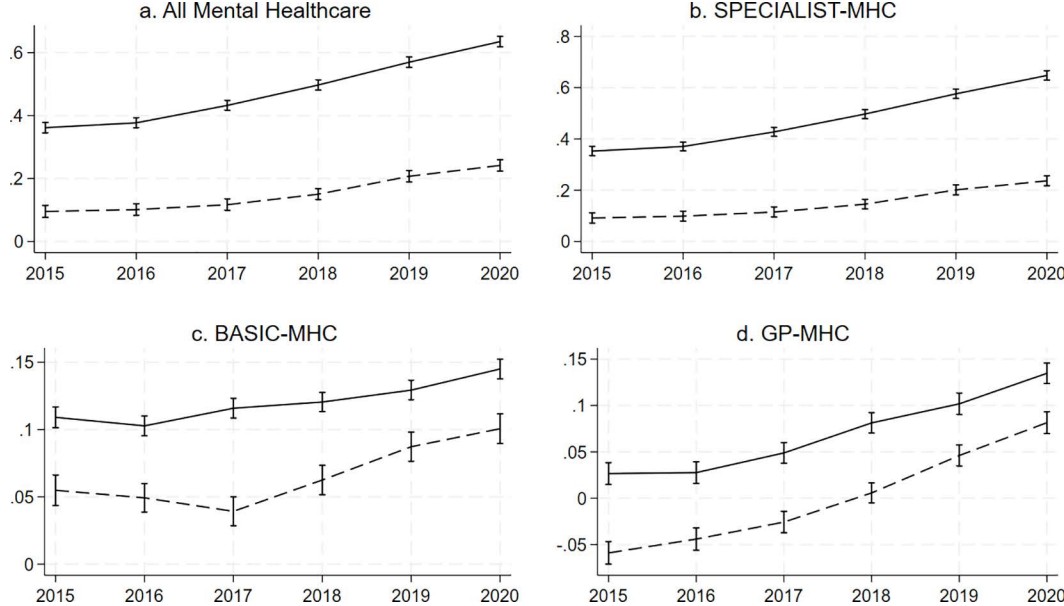

**Fig 4. Margins plots of the association between young age and costs by year and sex accounting for urbanicity, socioeconomic status, and the interaction between sex, age and year.** The figure shows graphs plotting the marginal estimates of the association between costs and young age by year and sex for all mental healthcare (a.) and by type of mental healthcare (b., c., d.). The Y-axis shows the regression coefficient for age. The X-axis shows the calendar year. The uninterrupted line represents women and the interrupted line represents men. The vertical bars intersecting the lines at each year represent confidence intervals.

(95%-CI = 0.63; 0.67) in 2020, while for men, it went from 0.09 (95%-CI = 0.07; 0.11) in 2015 to 0.24 (95%-CI = 0.22; 0.26) in 2020. For BASIC-MHC, the marginal estimate for the association with age and costs for women went from 0.11 (95%-CI = 0.10; 0.12) in 2015 to 0.15 (95%-CI = 0.14; 0.15) in 2020, while for men, it went from 0.05 (95%-CI = 0.04; 0.07) in 2015 to 0.10 (95%-CI = 0.09; 0.11) in 2020. For GP-MHC, this was from 0.03 (95%-CI = 0.01; 0.04) in 2015 to 0.13 (95%-CI = 0.12; 0.15) in 2020 and from −0.06 (95%-CI = −0.07; −0.05) in 2015 to 0.08 (95%-CI = 0.07; 0.09) in 2020 for women and men, respectively. The supplementary information contains an overview of the plotted marginal estimates and of the full regression outputs (S21-S28 in S1 File).

## Discussion

To our knowledge, this is the first study that examined patterns in the type of mental healthcare used in the Netherlands over time. Our findings showed that the association between young age and mental healthcare expenditure increased at the 4-number postal code level across all levels of the mental healthcare system over the period of 2015–2020. This rise in the association between young age and mental healthcare expenditure was strongest for the SPECIALIST-MHC. Regarding sex, the apparent rise in the association between young age and mental healthcare expenditure was stronger for women in total mental healthcare expenditure and SPECIALIST-MHC but stronger for men in BASIC-MHC and GP-MHC.

Our finding that the rise in the association between costs and age was stronger in women was expected. However, we did not hypothesize that this association would grow stronger for men in BASIC-MHC and GP-MHC over the observed period. This could be an indication that mental health is also worsening for young men, but only patients with mild problems end up in treatment at the BASIC-MHC and GP-MHC, as these mostly treat less severe problems than SPECIALIST-MHC professionals. This is in line with reports showing that over the period of 2015–2018 there was a larger increase in people using BASIC-MHC (11%) and GP-MHC (25%) compared to those using SPECIALIST-MHC (3%) [20,21]. One possible explanation is that young men's attitudes toward seeking mental healthcare are changing, making them more likely to seek help for milder complaints than older men. Help-seeking being seen as incompatible with male-gender roles has been identified as a barrier to help-seeking among men [22,23]. Changing male-gender role attributes shaped by society could contribute to a shift in help-seeking behaviour. An alternative explanation could be that men tend to show more externalizing symptoms, such as anger, delinquent behaviour, and substance use, in response to distress [24–28]. It could thus be the case that young men with more severe symptoms receive care less often due to limited capacity for treating addiction and externalizing symptoms in the Netherlands [29–31]. The number of treatments for addiction has remained relatively stable in recent years, but this may be due to health insurers imposing spendings caps on addiction care [32]. It could therefore be the case that BASIC-MHC and GP-MHC treat cases that would otherwise be treated by addiction care, which falls under SPECIALIST-MHC.

Our findings regarding gender could prompt clinicians to evaluate whether the care they offer to young people differs based on gender. The lower expenditures among young men, in particular, could have implications for policies aimed at improving accessibility and engagement with care for this group.

We have replicated our previous findings at a more granular postal code level that there is an increase in the association between young age and mental healthcare expenditures over time [5]. This finding could serve as an economic incentive for policymakers to invest in prevention strategies and address the underlying causes of mental distress in youth. The increasing association between young age and poor mental health is in line with results from a panel survey in Australia highlighting the generational aspect and cohort effects of deteriorating mental health in the population [33]. This study showed that people born in the 1980s and 1990s experience deteriorating mental health compared to previous generations [33]. Assuming the increased mental healthcare expenditure we found among young adults in the Dutch context is linked to poorer mental health in these generations, this is a worrying trend. Especially since we observed this most

prominently in SPECIALIST-MHC, which treats patients with more severe problems. Poor mental health is linked to poor physical health, and there is evidence that mental health conditions earlier in life are at the start of a cascade of health conditions later in life [34–36]. An increase in mental ill-health among young adults in recent years could thus also forecast poor physical health in this generation as they age. Our findings add to the concern that the current generation of young adults will suffer from a double morbidity of both mental and physical poor health while having to carry an increasingly large demographic burden as Dutch society as a whole ages [37]. It is also possible that the observed increase in mental healthcare costs among younger adults reflects greater help-seeking behaviour. This is supported by studies showing more positive attitudes toward mental health among younger people and a rise in the number of young individuals accessing services over time [38–40]. Additionally, mental distress may have become increasingly medicalized in recent years, which could have contributed to younger generations being more open to using mental health services compared to older generations [41–43].

### Strengths and limitations

We have previously described the limitations of working with insurance data at the postal code level [5]. In short, these are the ecological nature of the study not allowing for inferences at the individual level, costs not necessarily being the same as disease burden or patient numbers, and the exclusion of marginalised groups that fall outside the system of insurance. As we had access to 4-number postal code level data for this work, we could however adjust for neighbourhood-level socioeconomic status, which is an improvement from our previous work. Another major limitation is that we do not have the most recent data, meaning we could not examine how the found trends developed until the present. The lack of data on the period before 2015 meant we could not ascertain whether the overall higher mental healthcare costs in youth are a new phenomenon. *We* also did not apply sensitivity analyses to correct for missing data, but since there was very little missing data, its impact is likely negligible. Costs associated with forensic care, long-term clinical stays, and costs for social care and minors which are paid for by the municipality, were not included in our analysis, making our study an incomplete overview of trends in spending within the total mental healthcare landscape. Another issue is that we did not correct for inflation, meaning that our calculated prediction of log-transformed standardised costs (as shown in Figs 1 and 3) could in part have been influenced by inflation. However, the marginal estimate of the association between younger age and costs (as shown in Figs 2 and 4) is not influenced by this, as these reflect the proportion of mental healthcare costs made by younger people. Finally, although our results may provide some insight into predicted trends in countries with similar mental healthcare systems, they are not generalizable to other countries, as they are specific to the mental healthcare system in the Netherlands. Research in other countries is needed to better understand whether these trends are applicable globally and whether factors unique to countries explain differences.

The main strength of our study was the comprehensiveness of the sample, which comprised nearly the whole Dutch population. Our study also corrected for neighbourhood socioeconomic status and urbanicity, to prevent different age distributions across neighbourhoods of different levels of urbanicity and socioeconomic status from confounding the result.

### Conclusion

This study adds to the knowledge base on deteriorating youth mental health. While there have been increased expenditures in the more costly specialized mental healthcare for young women in particular, young men showed a stronger rise in basic and general practitioner mental healthcare. Our findings regarding young men are novel for the Netherlands, as most focus in the public debate has been on young women [6,8,44]. Further research should explore whether this trend is applicable in other countries, whether this trend is also due to an increased need for care in young men and whether the mental healthcare offered in the Netherlands sufficiently caters to all genders' needs.

## Supporting information

**S1 File. The online version contains supplementary information S1–S28.**
(PDF)

## Acknowledgments

We would like to thank Vektis for sharing the necessary data for this project with us.

## Author contributions

**Conceptualization:** Lotte Dijkstra, Jim van Os.

**Data curation:** Jim van Os.

**Formal analysis:** Lotte Dijkstra.

**Methodology:** Lotte Dijkstra, Jim van Os.

**Resources:** Jim van Os.

**Supervision:** Albert Batalla.

**Visualization:** Lotte Dijkstra.

**Writing – original draft:** Lotte Dijkstra.

**Writing – review & editing:** Sinan Gülöksüz, Albert Batalla, Jim van Os.

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
