## [Decision Letter · Decision Letter 0]

22 May 2025

PONE-D-25-20571
Mental healthcare expenditure among adults by type of mental healthcare and its association with age and sex in the Netherlands between 2015 and 2020
PLOS ONE

Dear Dr. Dijkstra

Thank you for submitting your manuscript to PLOS ONE. After careful consideration, we feel that it has merit but does not fully meet PLOS ONE’s publication criteria as it currently stands. Therefore, we invite you to submit a revised version of the manuscript that addresses the points raised during the review process.

We look forward to receiving your revised manuscript.

Kind regards,

Mu-Hong Chen, M.D., Ph.D.

Academic Editor

PLOS ONE

Journal Requirements:

3. In the online submission form you indicate that your data is not available for proprietary reasons and have provided a contact point for accessing this data. Please note that your current contact point is a co-author on this manuscript. According to our Data Policy, the contact point must not be an author on the manuscript and must be an institutional contact, ideally not an individual. Please revise your data statement to a non-author institutional point of contact, such as a data access or ethics committee, and send this to us via return email. Please also include contact information for the third party organization, and please include the full citation of where the data can be found.

Reviewers' comments:

Reviewer's Responses to Questions

**Comments to the Author**

1. Is the manuscript technically sound, and do the data support the conclusions?

Reviewer #1: Yes

Reviewer #2: Yes

2. Has the statistical analysis been performed appropriately and rigorously? 

Reviewer #1: Yes

Reviewer #2: Yes

3. Have the authors made all data underlying the findings in their manuscript fully available?

Reviewer #1: Yes

Reviewer #2: Yes

4. Is the manuscript presented in an intelligible fashion and written in standard English?

Reviewer #1: Yes

Reviewer #2: Yes

5. Review Comments to the Author

Reviewer #1: The authors utilized a health insurance database linked with postal codes and data from the Central Bureau for Statistics to estimate mental health costs in the Netherlands. They report that younger people incur higher costs than the 35–65-year-old group. The handling of missing data and covariates was generally reasonable. However, I have a few comments and questions I would like to raise:

1. Given that the database provides age information subdivided into groups (18–20 years and 5-year intervals thereafter), why did the analysis use a dichotomized age variable with a cut-off at 35 years? Would using 5- or 10-year age groups allow for a more detailed understanding of age-related differences in mental healthcare costs?

2. Were younger people already using more mental health resources before 2015, or was there a crossover in usage patterns at some point during the study period?

3. I noticed that mental health costs have increased year by year, but has the [number of visits or usage] by the public also increased annually? Or does this upward trend simply reflect increases in the cost of living or general inflation? How does this trend compare to that of other medical services?

4. Is it possible to provide information on the possible reasons for using mental health services, such as substance abuse, alcohol-related problems, emotional issues, self-harm, or psychosis? I know the authors discussed many possible reasons in the Discussion section, but is it possible to extract such information from the database itself?

5. The authors mention in the limitation section that “[Another major limitation is that we do not have the most recent data, meaning we could not examine how the found trends developed until the present.]” This is indeed unfortunate, as the period from 2020 to 2022 saw the global impact of COVID-19, during which the mental health system likely underwent significant changes.

Reviewer #2: This is a well-executed study using large-scale national data to explore important trends in mental healthcare use. The analysis is methodologically solid, and the findings have clear relevance for policy and planning. I have three suggestions that may help strengthen the manuscript:

1. In the Methods section, the choice to dichotomize age into 18–34 and 35–64 simplifies the analysis but may limit interpretability. This grouping could mask important differences within the broad 18–34 range, such as between individuals in early versus late young adulthood. It would strengthen the methodological transparency to explain the rationale behind this cutoff, and if data allow, to explore whether narrower age categories yield different or more detailed patterns in expenditure.

2. In the Discussion section, the manuscript effectively reports statistically significant differences in mental healthcare expenditures across groups. However, it remains unclear what the practical implications of these differences are from a clinical or policy perspective. Adding interpretation on the practical significance of your findings would enhance the utility of the results for policymakers and clinicians.

3. In the Discussion section, the authors interpret the rising mental healthcare expenditures among younger adults as indicative of worsening mental health in this group. While this is a reasonable and important interpretation, I wonder if the authors have also considered the possibility that the increase reflects a greater willingness to seek help, possibly due to reduced stigma or increased awareness. A brief acknowledgment of this alternative explanation, along with any supporting literature if available, could offer a more nuanced perspective.

6. PLOS authors have the option to publish the peer review history of their article (what does this mean?). If published, this will include your full peer review and any attached files.

Reviewer #1: No

Reviewer #2: No

---

## [Author Response · Author response to Decision Letter 1]

9 Jul 2025

Dear reviewers,

We thank you for having taken the time to revise our manuscript. We have attached a response to your comments which we hope is to your satisfaction. If you have further questions we look forward to hearing them.

Kind regards,

Lotte Dijkstra

---

## [Decision Letter · Decision Letter 1]

31 Jul 2025

Mental healthcare expenditure among adults by type of mental healthcare and its association with age and sex in the Netherlands between 2015 and 2020

PONE-D-25-20571R1

Dear Dr. Lotte Dijkstra,

We’re pleased to inform you that your manuscript has been judged scientifically suitable for publication and will be formally accepted for publication once it meets all outstanding technical requirements.

Kind regards,

Mu-Hong Chen, M.D., Ph.D.

Academic Editor

PLOS ONE

Additional Editor Comments (optional):

Reviewers' comments:

Reviewer's Responses to Questions

**Comments to the Author**

1. If the authors have adequately addressed your comments raised in a previous round of review and you feel that this manuscript is now acceptable for publication, you may indicate that here to bypass the “Comments to the Author” section, enter your conflict of interest statement in the “Confidential to Editor” section, and submit your "Accept" recommendation.

Reviewer #1: All comments have been addressed

Reviewer #2: All comments have been addressed

2. Is the manuscript technically sound, and do the data support the conclusions?

Reviewer #1: Yes

Reviewer #2: Yes

3. Has the statistical analysis been performed appropriately and rigorously? 

Reviewer #1: Yes

Reviewer #2: Yes

4. Have the authors made all data underlying the findings in their manuscript fully available?

Reviewer #1: Yes

Reviewer #2: Yes

5. Is the manuscript presented in an intelligible fashion and written in standard English?

Reviewer #1: Yes

Reviewer #2: Yes

6. Review Comments to the Author

Reviewer #1: All comments are addressed. I have no further comment.

This study, despite its several limitations, is worthy of publication after revision.

Reviewer #2: The authors have provided satisfactory responses to the previously raised points, and the revised manuscript reflects the requested clarifications. I have no additional substantive comments at this stage.

7. PLOS authors have the option to publish the peer review history of their article (what does this mean?). If published, this will include your full peer review and any attached files.

Reviewer #1: No

Reviewer #2: No

---

## [Editor Report · Acceptance letter]

PONE-D-25-20571R1

PLOS ONE

Dear Dr. Dijkstra,

I'm pleased to inform you that your manuscript has been deemed suitable for publication in PLOS ONE. Congratulations! Your manuscript is now being handed over to our production team.

Kind regards,

on behalf of

Dr. Mu-Hong Chen

Academic Editor

PLOS ONE